# Unlocking Robust Segmentation Across All Age Groups via Continual Learning

**Chih-Ying Liu**[1]                                       YING1029@STANFORD.EDU
[1] *Stanford University*

**Jeya Maria Jose Valanarasu**[1]                          JMJOSE@STANFORD.EDU
**Camila Gonzalez**[1]                                     CAMGONZA@STANFORD.EDU
**Curtis Langlotz**[1]                                     LANGLOTZ@STANFORD.EDU
**Andrew Ng**[1]                                           ANG@CS.STANFORD.EDU
**Sergios Gatidis**[1]                                     SGATIDIS@STANFORD.EDU

**Editors:** Accepted for publication at MIDL 2024

## Abstract

Most deep learning models in medical imaging are trained on adult data with unclear performance on pediatric images. In this work, we aim to address this challenge in the context of automated anatomy segmentation in whole-body Computed Tomography (CT). We evaluate the performance of CT organ segmentation algorithms trained on adult data when applied to pediatric CT volumes and identify substantial age-dependent underperformance. We subsequently propose and evaluate strategies, including data augmentation and continual learning approaches, to achieve good segmentation accuracy across all age groups. Our best-performing model, trained using continual learning, achieves high segmentation accuracy on both adult and pediatric data (Dice scores of 0.90 and 0.84 respectively).

**Keywords:** Image Segmentation, Age bias, Continual Learning.

## 1. Introduction

Anatomy segmentation is a crucial part of numerous medical image processing pipelines (McBee et al., 2018). Current state-of-the-art models for organ segmentation on Computed Tomography (CT) are based on deep learning frameworks (Simpson et al., 2019; Jin et al., 2020; Feng et al., 2020; Wasserthal et al., 2023) and achieve excellent results when trained on representative adult training data. How these results transfer to a pediatric population is however unknown. Children have substantially different anatomic features compared to adults. So, segmentation models trained on adult data do not directly translate to pediatric populations. Reasons for under-representation of pediatric applications in medical imaging AI research include limited availability of accessible pediatric training data and lack of general awareness within the research community. It is important to address this challenge (Adamson et al., 2022) to improve the availability of emerging AI technologies for children.

The purpose of this work is to evaluate the performance of TotalSegmentator on pediatric CT and to propose strategies for optimization of model performance on pediatric data without sacrificing performance on adults, resulting in a model that produces robust predictions across age groups. To this end, we compare various approaches for domain adaptation and model fine-tuning, including continual learning methods (González et al., 2023), using publicly available pediatric (Jordan et al., 2022) and adult (Wasserthal et al., 2023) CT datasets with anatomic labels.

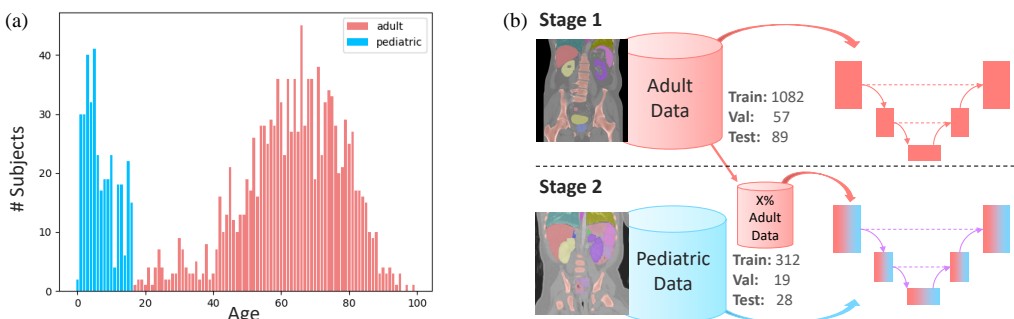

Figure 1: (a) Age distribution of datasets. (b) Continual Learning with Rehearsal.

## 2. Methods

**Dataset:** We use two public CT datasets for adult (Wasserthal et al., 2023) and pediatric (Jordan et al., 2022) patients. Figure 1(a) shows the age distribution and sizes of both datasets. We train our models on the 19 classes labeled in both datasets. We ensure balanced age representation across all data splits.

**Out-of-The-Box TotalSegmentator Model Performance on Pediatric Data:** To assess the performance of TotalSegmentator (*TS*) (Wasserthal et al., 2023) on pediatric data, inference was performed on the pediatric data set by matching and combining predictions into the 19 labels available in the pediatric dataset.

**Domain Adaptation via Augmentation:** In a first step, we investigated to which extent simple image preprocessing can improve model performance on pediatric data (*DA*). To this end, we up-scaled the spatial size of pediatric CT datasets by a factor of 1.5 before inference in an attempt to adjust for differences in body size between children and adults.

**Baseline Models:** We compare models trained on adult (*AdultSeg*), pediatric (*PediatricSeg*), and a mix of adult and pediatric data (*MixSeg*). We used the nn-UNet model framework (Isensee et al., 2018, 2021) with 1.5-mm isotropic resolution on 19 classes.

**Continual Learning:** We adopt rehearsal learning (*CL*), the simplest form of continual learning, to adapt to pediatric data while preserving performance on adult data. The model is trained on adult data in the first stage. Then, the model is further trained on pediatric plus $p$ percentage of adult data as illustrated in Figure 1(b). We also experiment with sequential learning (*Sequential*), where no adult data is sampled in the second stage.

**Metrics:** We assess the dice similarity coefficient (DSC) and normalized surface distance (NSD) between predicted segmentations and human annotations across age bins: 0-3, 4-6, 7-9, 10-12, and 13-16 for pediatric data and 17+ for adult data.

## 3. Results

Table 1 shows the DSC and NSD of each method across age bins. We consider the out-of-the-box model Total Segmentator and the model trained on adult dataset as two baselines.

Total Segmentator exhibits poor performance on young patients. Upscaling pediatric scans achieves 13% improvement on patients under three years old. Models that are trained on only adult or pediatric data underperform on the age ranges not seen during training. Training on a mix of both datasets produces a considerable improvement, yet rehearsal

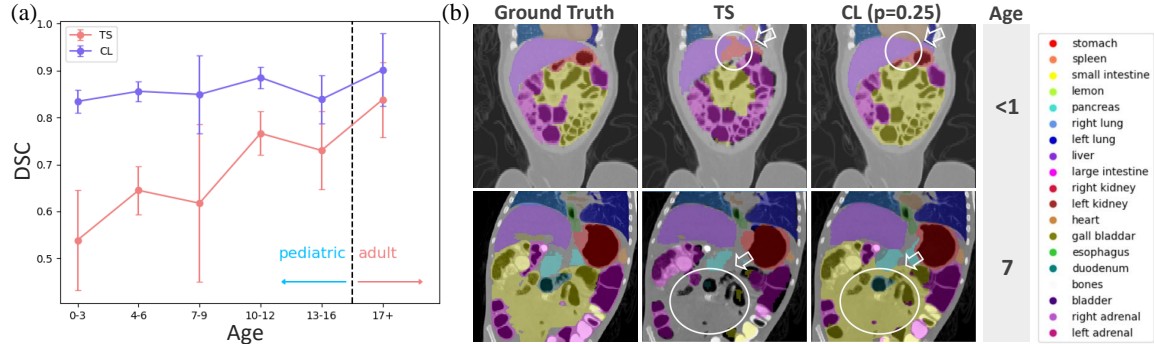

Figure 2: (a) Performance across age groups. (b) We observed clear segmentation errors using TotalSegmentator (TS) errors markedly improved using our proposed approach (CL).

learning - which only interleaves a portion of adult scans - performs best across most age groups. By adjusting the adult data sample percentage $p$, we can trade-off between adult and pediatric performance

Figure 2 shows the predictions and performance of Total Segmentator and rehearsal learning across all ages. Rehearsal learning with sample probability 0.25 achieves mean DSC 0.84 on pediatric data, and 0.90 on adult data. Compared the adult baseline model (*AdultSeg*), it achieves DSC improvement of 0.18 on pediatric data, with a minimal decrease of DSC on adult data of 0.01.

Table 1: Mean DSC and NSD of each methods across age bins. $A$ and $P$ represent adult and pediatric. TS = TotalSegmentator, CL=Continual Learning, DA=Domain Adaptation.

| Method | Network | Data | DSC/NSD (%) | | | | | |
|---|---|---|---|---|---|---|---|---|
| | | | Pediatric | | | | | Adult |
| | | | 0-3 | 4-6 | 7-9 | 10-12 | 13-16 | 17+ |
| TS | TS | A | 53.1/51.7 | 62.9/57.3 | 60.5/57.8 | 75.4/74.3 | 71.3/66.1 | 83.8/91.1 |
| DA | TS | A | 66.7/68.3 | 70.2/68.2 | 66.5/65.4 | 77.2/76.1 | 71.8/67.0 | −/− |
| AdultSeg | nn-UNet | A | 57.3/57.7 | 65.6/62.0 | 62.7/60.8 | 78.3/77.6 | 75.2/71.2 | **91.4/94.8** |
| PediatricSeg | nn-UNet | P | 81.8/86.6 | 83.2/85.5 | 85.3/88.1 | 86.6/89.3 | 81.3/80.3 | 72.2/73.2 |
| MixSeg | nn-UNet | A+P | 81.8/86.7 | 82.9/85.3 | 83.4/86.4 | 86.9/89.6 | 81.8/80.0 | 90.7/94.2 |
| Sequential | nn-UNet | A+P | 81.4/85.9 | 82.7/84.5 | 83.6/85.6 | 86.2/88.5 | 80.3/77.8 | 73.2/73.7 |
| CL(p=0.25) | nn-UNet | A+P | **83.0/88.6** | **84.7/87.7** | **85.0**/88.5 | **87.6/90.7** | **83.5/83.4** | 90.0/93.5 |
| CL(p=0.6) | nn-UNet | A+P | 82.7/87.9 | 83.8/86.7 | 85.0/**89.0** | 87.4/90.5 | 83.1/82.0 | 90.7/84.0 |
| CL(p=1.0) | nn-UNet | A+P | 81.5/86.7 | 82.8/85.1 | 83.3/85.8 | 87.3/90.1 | 81.3/80.2 | 90.6/94.1 |

## 4. Conclusion

We developed a model that performs well over all age groups by continual learning. Our model achieves a Dice score of 84.2% on pediatric data, and 90.0% on adult data. We have so far focused on the 19 coarse-graind class labels that have overlapping in the both datasets, and leave the experiments of fine-grained labels, that require additional annotation efforts, to future work.

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
