# OpenReview forum: "Unlocking Robust Segmentation Across All Age Groups via Continual Learning"
_MIDL.io/2024/Short_Papers — MIDL 2024 Short Papers_

### Official Review · Reviewer_hFRz · 2024-04-24

**Confidence:** 4
**Final Rating:** 4

**Review:**

The authors evaluate various segmentation strategies, in whole-body CT, applied to pediatric data. The authors essentially evaluate UNets trained in various ways, augmentation, and continual learning.

I think this is a really nice direction, and we need more of this sort of analysis, as pediatric data is often ignored in studies. It's a bit surprising to me that well-developed nnunets, with pediatric training data, do not perform better, and CL is necessary -- I would recommend that the authors work on this more in the future to understand this. I think to properly analyze this, a more rigorous experiment is needed leading to a more substantial paper, but I do believe that this is a nice analysis that should be discussed at MIDL already!

---

### Decision · Program_Chairs · 2024-04-26

Accept